# Regional variation in COVID-19 positive hospitalisation across Scotland during the first wave of the pandemic and its relation to population density: A cross-sectional observation study

**Andrew Rideout[1], Calum Murray[2], Chris Isles[3]***

1 Department of Public Health, Dumfries and Galloway Royal Infirmary, Dumfries, Scotland, 2 Education Centre, Dumfries and Galloway Royal Infirmary, Dumfries, Scotland, 3 Medical Unit, Dumfries and Galloway Royal Infirmary, Dumfries, Scotland

* christopher.isles@nhs.scot

**Data Availability Statement:** All relevant data are within the paper and its Supporting Information files.

## Abstract

### Background

There have been large regional differences in COVID-19 virus activity across the UK with many commentators suggesting that these are related to age, ethnicity and social class. There has also been a focus on cases, hospitalisations and deaths rather than on hospitalisation rates expressed per 100,000 population. The purpose of our study was to examine regional variation in COVID-19 positive hospitalisation rates in Scotland during the first wave of the pandemic and the possibility that these might be related to population density.

### Methods and findings

This was a repeated point prevalence study. The number of COVID-19 positive patients hospitalised in the eleven Scottish mainland health boards peaked at 1517 on 19th April, then fell to a low of 243 on 16th August before rising slightly to 262 on 15th September. In July, August and September only four boards had more than 5 hospitalised patients. There was a statistically significant relationship between hospitalisation rates and population density on 97.7% of individual days during the first wave of the pandemic (Pearson's r 0.62–0.93, with 123 of a possible 174 days having p values <0.001). Multiple linear regression analyses performed on data from the 11 mainland boards across six time points suggest that population density accounted for 70.2% of the variation in hospitalisation rate in April, 72.3% in May, 81.2% in June, 91.0% in July, 91.0% in August, and 88.1% in September. Neither population median age nor median social deprivation score at health board level were statistically significant in the final model for hospitalisation.

**Funding:** The authors received no specific funding for this work.

**Competing interests:** The authors have declared that no competing interests exist.

## Conclusion

There were large differences in crude COVID-19 hospitalisation rates across the 11 mainland Scottish health boards, that were significantly related to population density. Given that lockdown was originally introduced to prevent the NHS from being overwhelmed, we believe our results support a regional rather than a national approach to lifting or reimposing more restrictive measures, and that hospitalisation rates should be part of the decision making process.

## Introduction

While much has been made of the UK's COVID-19 death toll, which as of 19th May 2021 was seventeenth highest in the world when expressed per million population [1], less has been written about the determinants of regional variation in coronavirus activity within the UK, or of the local population and social factors required when planning a return towards normal activity in the recovery stage [2]. The possibility that population density might influence virus transmission rates and clinical outcome is of interest given the wide regional variations in COVID-19 positive cases seen throughout the United Kingdom.

Evidence thus far supports an association between population density and virus activity though it is not yet clear from the literature whether density per se is key or whether density is merely a proxy for more important social determinants [3–8]. It is also evident that the concentration of people within a densely populated area does not necessarily lead to high infection rates if cities adopt robust social distancing, mask wearing and contact tracing measures [9, 10].

Scotland's population was 5,463,300 in mid-2019 and average population density in Scotland was 70/sq km that year. There are large differences in population density between mainland boards ranging from 10/sq km in Highland, the least densely populated board, to 1072/sq km in Greater Glasgow and Clyde, the most densely populated board [11] (Fig 1). For comparison, average population densities for the rest of the United Kingdom are as follows: England (432/sq km), Wales (152/sq km) and Northern Ireland (137/sq km) [12]. It is against this background that we undertook an analysis of hospitalization of COVID-19 positive patients in Scotland during the first wave of the pandemic. We were particularly interested in the relation between hospitalisation rates, expressed per 100,000 population, and population density.

## Methods

We used Scottish Government data to identify the number of hospitalised COVID-19 positive patients in all eleven Scottish mainland health boards at midnight each night from 26th March to 15th September 2020 [13] and mid 2019 population estimates for Scotland to calculate hospitalisation rates/100,000 population [11]. Our analysis was in three parts. First, we compared crude hospitalisation rates across the 11 Scottish mainland boards. Second we examined the relationship between crude hospitalisation rate (per 100,000 population) and population density (population per $km^2$) for every day (26th March 2020 to 15th September 2020) of the 'first wave' of the UK COVID-19 pandemic across the 11 Scottish mainland health boards. Third, we undertook stepwise multiple linear regression analysis with backward elimination in order to determine the effects of age and deprivation on hospital admissions. For descriptive purposes, we selected six time points to illustrate these analyses: 19th April, 15th May, 14th June, 19th July, 16th August and 15th September. 19th April was the peak of the pandemic in

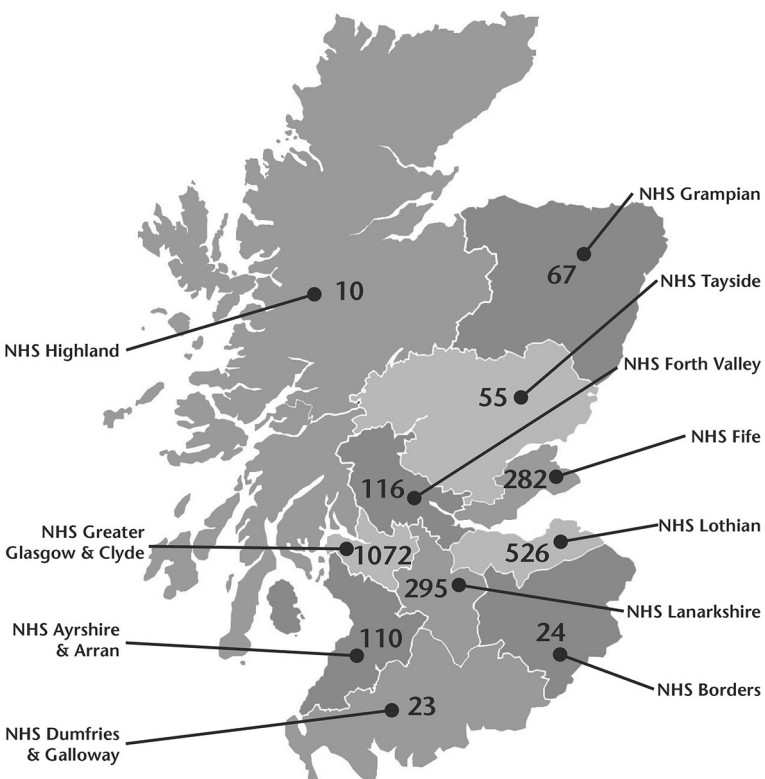

**Fig 1. Map of Scotland showing population densities of the 11 mainland health boards.**

Scotland, 14th June was the week hospitalised cases dropped below 5 in five of the 11 Scottish health boards, 16th August was the day of the lowest number of COVID-19 positive hospitalisations in Scotland and 15th September was during the week when the prime minister confirmed that we were beginning to see a second wave of infections [14]. 15th May and 19th July were approximately midway between April-June and June-August time points.

When numbers of COVID-19 positive patients in Scottish hospitals drop below 5 per health board the Scottish Government report these as 'less than 5' in order to reduce the risk that patients might be identified. We were able to determine exact numbers of hospitalised patients in Dumfries and Galloway but not the other health boards and opted to allocate a 'two' whenever these health boards reported less than five inpatients. We chose not to analyse the number of COVID-19 positive cases in each health board because of the risk of bias due to different testing regimens. Nor did we attempt to compare death rates. This was because some deaths recorded as due to COVID-19 are possible or presumed COVID-19 in patients who have not tested positive for the virus (clinical diagnosis rather than laboratory diagnosis); because not everyone who died with COVID-19 symptoms was tested for the virus, particularly during the early phase of the pandemic; because not everyone who died within 28 days of testing positive for COVID-19 has died of a COVID-19 related illness, and because deaths make up only a small part of total COVID-19 cases, exacerbating small number effects in calculations.

## Statistical analyses

We compared crude hospitalisation rates between health boards by chi square test and odds ratios with 95% confidence intervals where appropriate and correlated hospitalisation rates with

population density using Pearson's correlation coefficient. Crude hospitalisation rates, population proportions, confidence intervals, and chi square test were calculated manually, and odds ratios were calculated using Epi Info Statcalc (v7.2.3.1) [15]. Pearson's correlation coefficients and regression analyses were calculated using JASP Version 0.10.2 [16]. Three covariates were included in the stepwise multiple linear regression analysis: population density, median age, and median deprivation decile (Scottish Index of Multiple Deprivation (SIMD) decile) [17] for each health board. We calculated median SIMD decile for each health board in Scotland as a measure of population experience of deprivation, where SIMD1 represents the 10% of the Scottish population living in greatest deprivation, and SIMD10 the 10% living in the least deprivation. The regression equation was calculated to predict hospitalisation rate based on population density, age and deprivation at all six time points and for all eleven mainland health boards.

### Patient and public involvement

No patient or member of the public was involved in the design, analysis or reporting of this study.

## Results

Table 1 shows population and population density for each of the 11 mainland health boards together with numbers hospitalised and crude COVID-19 positive hospitalisation rates per 100,000 population on each of the six time points, while Fig 2 shows daily hospitalisation rates for COVID-19 positive patients per 100,000 population in four of these health boards for illustrative purposes: Dumfries and Galloway, a rural health board which has a border with England; Ayrshire and Arran, a neighbouring health board; Highland, the least densely populated mainland health board; and Greater Glasgow & Clyde, the largest and most densely populated board in Scotland. Daily hospitalisation rates for all 11 mainland boards from 26th March to 15th September are available in a S1 File.

The number of COVID-19 positive patients hospitalised in the eleven Scottish mainland health boards was 1517 on 19th April, 1062 on 15th May, 575 on 14th June and 302 on 19th July. Numbers fell to a low of 243 on 16th August before rising slightly to 262 on 15th September. Crude hospitalisation rates fell in 10 of 11 health boards between 19th April and 15th May, and between 15th May and 14th June. On 19th July, 16th August and 15th September only four of Scotland's mainland health boards, namely Greater Glasgow and Clyde, Lothian, Grampian and Lanarkshire, had more than five hospitalised COVID-19 positive patients. Greater Glasgow and Clyde, the health board with the highest population density, had the highest hospitalisation rates throughout the first wave of the pandemic (Fig 2). Significant inter board variations were seen at each of the time points shown in Table 1 (chi square test (df10) p <0.001 on all dates).

Fig 3 shows the relationship between hospitalisation rates and population density on 14th June for illustrative purposes. This relationship was statistically significant on all except four days (26th, 29th and 31st March, and 1st April) during the first wave of the pandemic (Pearson's correlation coefficient was in the range 0.62–0.93 with 13 p values between 0.05 and 0.01, 34 p values between 0.01 and 0.001, 123 p values <0.001). This was largely driven by increased hospitalisation in Greater Glasgow and Clyde Health Board, which is not only the largest but also the most densely populated Health Board and the Health Board with one of the greater levels of deprivation. When Greater Glasgow and Clyde was removed from the analysis a statistically significant correlation between population density and hospitalisation rates was demonstrated on 115/174 days (66.1% of occasions) during the first wave of the pandemic.

Multiple linear regression analyses performed on data from the 11 mainland boards across the six time points showed that the independent variables of median age and median deprivation

**Table 1. Population, population density and confirmed COVID-19 hospitalisation rates/100,000 population at six time points during the first wave of the pandemic.**

| Health Board | Population Mid June 2019 (population density/km²) | 19th April cases in hospital | Hospital-isations/ 100,000 (95% CI) | 15th May cases in hospital | Hospital-isations/ 100,000 (95% CI) | 14th June cases in hospital | Hospital-isations/ 100,000 (95% CI) | 19th July cases in hospital | Hospital-isations/ 100,000 (95% CI) | 16th August cases in hospital | Hospital-isations/ 100,000 (95% CI) | 15th September cases in hospital | Hospital-isations/ 100,000 (95% CI) |
|---|---|---|---|---|---|---|---|---|---|---|---|---|---|
| Ayrshire & Arran | 369,360 (110) | 91 | 24.6 (19.6, 29.7) | 42 | 11.4 (7.9, 14.8) | 5 | 1.4 (0.2, 2.5) | 2* | 0.5 (0.0,1.3) | 2* | 0.5 (0.0,1.3) | 2* | 0.5 (0.0,1.3) |
| Borders | 115,510 (24) | 43 | 37.2 (26.1, 48.4) | 25 | 21.6 (13.2, 30.1) | 2* | 1.7 (0.0,4.1) | 2* | 1.7 (0.0,4.1) | 2* | 1.7 (0.0,4.1) | 2* | 1.7 (0.0,4.1) |
| Dumfries & Galloway | 148,860 (23) | 20 | 13.4 (7.5, 19.3) | 2 | 1.3 (0.0, 3.2) | 0 | 0.00 (0.0, 0.0) | 0 | 0.0 (0.0, 0.0) | 1 | 0.7 (0.0, 2.0) | 3 | 0.8 (0.0, 1.7) |
| Fife | 373,550 (282) | 105 | 28.1 (22.7, 33.5) | 75 | 20.1 (15.5, 24.6) | 57 | 15.3 (11.3, 19.2) | 2* | 0.5 (0.0, 1.3) | 2* | 0.5 (0.0, 1.3) | 2* | 0.5 (0.0, 1.3) |
| Forth Valley | 306,640 (116) | 50 | 16.3 (11.8, 20.8) | 21 | 6.8 (3.9, 9.8) | 5 | 1.6 (0.2, 3.1) | 2* | 0.7 (0.0, 1.6) | 2* | 0.7 (0.0, 1.6) | 2* | 0.7 (0.0, 1.6) |
| Grampian | 585,700 (67) | 72 | 12.3 (9.5, 15.1) | 96 | 16.4 (13.1, 19.7) | 56 | 9.6 (7.1, 12.1) | 29 | 5.0 (3.1, 6.8) | 16 | 2.7 (1.4, 4.1) | 20 | 3.4 (1.9, 4.9) |
| Greater Glasgow & Clyde | 1,183,120 (1072) | 593 | 50.1 (46.1, 54.2) | 460 | 38.9 (35.3, 42.2) | 255 | 21.6 (18.9, 24.2) | 171 | 14.5 (12.3, 16.6) | 135 | 11.4 (9.5, 13.3) | 138 | 11.6 (9.7, 13.6) |
| Highland | 321,700 (10) | 45 | 14.0 (9.9, 18.1) | 5 | 1.6 (0.2, 2.9) | 7 | 2.2 (0.6, 3.8) | 2* | 0.6 (0.0, 1.5) | 2* | 0.6 (0.0, 1.5) | 2* | 0.6 (0.0, 1.5) |
| Lanarkshire | 661,900 (295) | 180 | 27.2 (23.2, 31.2) | 113 | 17.1 (14.9, 20.2) | 49 | 7.4 (5.3, 9.5) | 19 | 2.9 (1.6, 4.2) | 6 | 0.9 (0.2, 1.6) | 16 | 2.4 (1.2, 3.6) |
| Lothian | 907,580 (526) | 228 | 25.1 (21.9, 28.4) | 200 | 22.0 (19.0, 25.1) | 134 | 14.8 (12.3, 17.3) | 78 | 8.6 (6.7, 10.5) | 81 | 8.9 (7.0, 10.9) | 78 | 8.6 (6.7, 10.5) |
| Tayside | 417,470 (55) | 90 | 21.6 (17.0, 26.0) | 23 | 5.5 (3.3, 7.8) | 6 | 1.4 (0.3, 2.6) | 2* | 0.5 (0.0, 1.1) | 2* | 0.5 (0.0, 1.1) | 2* | 0.5 (0.0, 1.1) |
| Scotland | 5,463,300 (70) | 1520 | 27.8 (26.4, 29.2) | 1066 | 19.5 (18.3, 20.7) | 575 | 10.5 (9.7, 11.4) | 302 | 5.5 (4.9, 6.2) | 243 | 4.4 (3.9, 5.0) | 262 | 4.8 (4.2, 5.4) |

Footnote Table 1: Scottish Government suppressed counts on their website below 5, so a proxy count of 2 cases was used for these dates when true count was not known.

score were not statistically significant in the final model for hospitalisations, confirming that the most significant driver for hospitalisation identified within this study was population density. Significant regression equations were found for 19th April ($F_{(2,8)}$ = 9.409, p = 0.008), with an $R^2$ of 0.702; for 15th May ($F_{(1,9)}$ = 21.980, p = 0.001), with an $R^2$ of 0.709; for 14th June ($F_{(1,9)}$ = 29.01, p < .001), with an $R^2$ of 0.763; for 19th July ($F_{(2,8)}$ = 40.15, p<0.001) with an $R^2$ of 0.909, for 16th August ($F_{(2,8)}$ = 35.63, p<0.001) with an $R^2$ of 0.899, and for 15th September ($F_{(2,8)}$ = 27.57, p<0.001) with an $R^2$ of 0.873. These data suggest that population density accounted for 70.2% of the variation in hospitalisation rate in April, 70.9% in May, 76.3% in June, 90.9% in July, 89.9% in August, and 87.3% in September. The models are not an exact fit (suggesting that there are other factors at play), but provide a reasonable approximation to the observed numbers of hospitalisations in each of the eleven health boards at these six time points.

Data released by the Scottish Government confirm there was sufficient hospital bed capacity across Scotland during the first wave of the pandemic [18].

## Discussion

The results of our survey show significant regional variations in hospitalisation rates for COVID-19 positive patients during the first wave of the COVID-19 pandemic in Scotland.

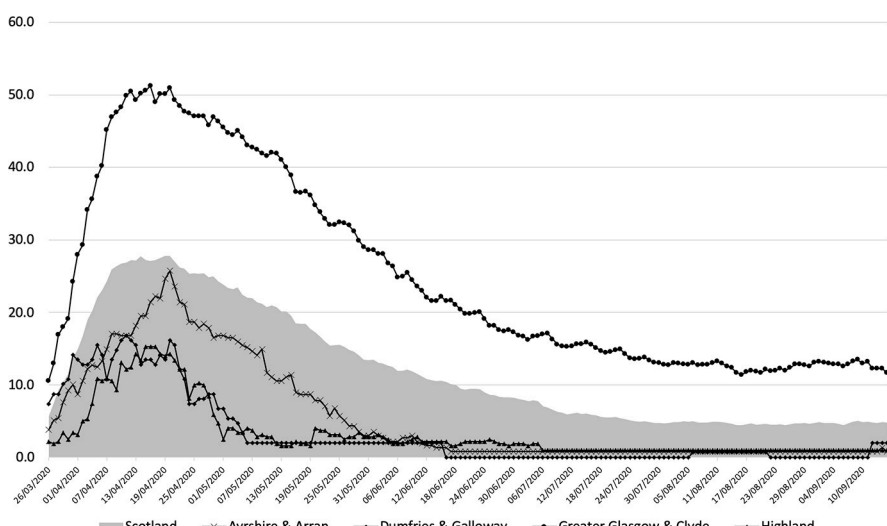

**Fig 2. Daily hospitalisation rates for COVID-19 positive patients/100,000 population in Scotland, Ayrshire & Arran, Dumfries and Galloway, Greater Glasgow and Clyde and Highland (shaded).**

These variations were significantly associated with health board population density. The relationship was one of increasing hospitalisation rate being positively associated with greater population density. This was independent of population median age and median social deprivation score at health board level. Hospitalisation peaked 2–3 weeks after lockdown and fell steadily to a low on 16th August. At no point during the first wave of the pandemic was the NHS overwhelmed in Scotland.

Population density could have contributed to the variation in hospitalisation for the following three reasons. First, a low population density may make it easier to practise social distancing. Second, remote and rural areas such as Highland, Borders and Dumfries & Galloway lack

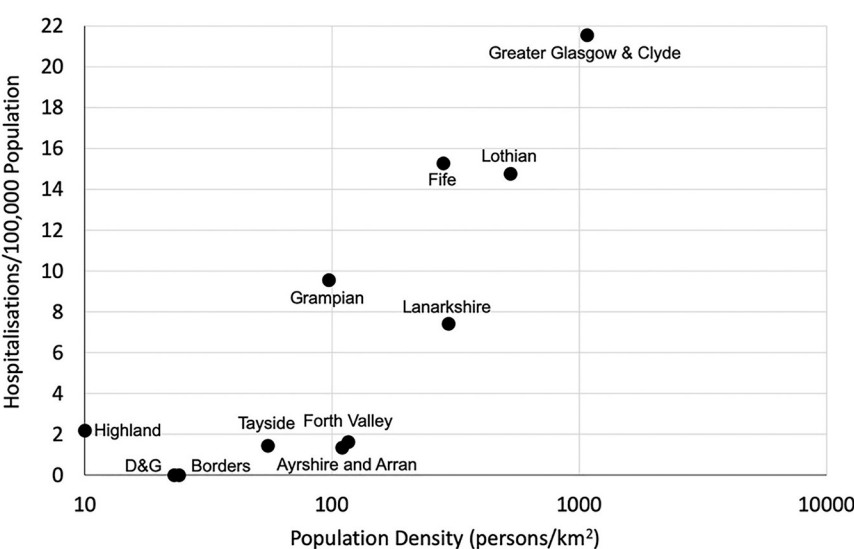

**Fig 3. Confirmed COVID-19 hospitalisation rates/100,000 versus population density in all 11 Scottish mainland health boards on 14th June 2020.** Similar graphs showing data for the five other time points are available on request. The graph is shown with a logarithmic axis for clarity.

the connectivity (namely the large office buildings, socialisation spaces, and multiple modes of public transport) that draws people into close proximity [5]. Third, rural communities have been observed to behave distinctively in times of natural disaster by expressing greater social control, thereby ensuring compliance with lockdown [19]. We acknowledge, however, that population density cannot be the only explanation [9, 10] and recognise the possibility that density may merely be a proxy for other social determinants [4–6, 8]

If we accept that we are unlikely to abolish risk completely, then we believe it should be possible to relax restrictions earlier in those regions of the UK with lower virus activity when it is judged that the risks of lockdown [20, 21] outweigh risks of the virus, when there is a test, trace and isolate system in place and particularly now that a vaccination programme is well underway. Whilst Reproductive Number ($R_0$) remains the gold standard for measuring the spread of an epidemic, in practice it is based on a number of assumptions and is complex to calculate. An additional measure could be hospitalisation rates within a region, which allow daily small area monitoring of disease spread, albeit with a longer delay caused by the lag between transmission, onset of clinical disease and deterioration to the point of requiring hospitalisation. Other high income countries, including Japan, Germany, South Korea and Hong Kong, have adopted a regional rather than a national approach to lifting or reimposing restrictions based on the number of new cases/100,000 per week [22]. Our data support this approach, which was also the approach taken by the Scottish government during the first wave of the pandemic. Given the extreme pressure on hospital beds during the second wave, we suggest that hospitalisation rates expressed per 100,000 population could be a useful additional metric when deciding whether to lift or reimpose restrictions.

## Strengths and limitations

All emergency health care in the UK is free at the point of delivery, allowing equal access for all who require access to medical evaluation and potential hospitalisation. It was never our intention to examine in depth the reasons behind the differences in hospitalisation, nor could we have ever hoped to do so with the dataset to which we had access. We are not trying to infer that hospitalisation rates are purely a function of population density, more that hospitalisation reflects the ability of the NHS to cope with the COVID-19 pandemic. Hospitalisation is likely to be related to other factors such as age, ethnicity, comorbidity, health inequalities and personal behaviours in addition to the characteristics of the circulating virus. At a system wide level, we have shown that population density appears to be a larger driver of need for hospital beds than age and social deprivation, but are unable to determine the relative contributions of ethnicity and comorbidity. We recognise that SIMD identifies deprived areas within Scotland rather than deprived individuals and would anticipate higher burden of infection and hospitalisation in communities with higher levels of socio-economic deprivation. We acknowledge these issues as limitations and suggest they could form the basis for further research.

## Conclusions

We have confirmed there were large differences in COVID-19 hospitalisation rates across the 11 mainland Scottish health boards during the first wave of the COVID-19 pandemic. At Health Board level these were significantly related to population density but not to age, social deprivation, or hospital bed capacity, which did not impose an artificial barrier to hospital admission based on clinical need in Scotland during the first wave of the pandemic. Based on these data, and the premise that lockdown was originally introduced to prevent the NHS from being overwhelmed, we believe our results support a regional rather than a national approach to lifting or reimposing more restrictive measures, and that hospitalisation rates should be part of the decision making process.

## Supporting information

**S1 Checklist.**
(DOCX)

**S1 File.**
(XLSX)

## Author Contributions

**Conceptualization:** Chris Isles.

**Data curation:** Calum Murray.

**Formal analysis:** Andrew Rideout.

**Writing – original draft:** Chris Isles.

**Writing – review & editing:** Andrew Rideout, Calum Murray, Chris Isles.

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
