## [Decision Letter · Decision Letter 0]

11 May 2021

PONE-D-20-38210

Regional variation in COVID-19 positive hospitalisation across Scotland during the first wave of the pandemic and its relation to population density: a cross-sectional observation study.

PLOS ONE

Dear Dr. Isles,

Thank you for submitting your manuscript to PLOS ONE. After careful consideration, we feel that it has merit but does not fully meet PLOS ONE’s publication criteria as it currently stands. Therefore, we invite you to submit a revised version of the manuscript that addresses the points raised during the review process.

We look forward to receiving your revised manuscript.

Kind regards,

Karyn Morrissey

Academic Editor

PLOS ONE

Journal Requirements:

3.We note that Figure(s) 1 in your submission contain map  images which may be copyrighted. All PLOS content is published under the Creative Commons Attribution License (CC BY 4.0), which means that the manuscript, images, and Supporting Information files will be freely available online, and any third party is permitted to access, download, copy, distribute, and use these materials in any way, even commercially, with proper attribution. For these reasons, we cannot publish previously copyrighted maps or satellite images created using proprietary data, such as Google software (Google Maps, Street View, and Earth). For more information, see our copyright guidelines: http://journals.plos.org/plosone/s/licenses-and-copyright.

a) You may seek permission from the original copyright holder of Figure(s) 1 to publish the content specifically under the CC BY 4.0 license. 

Reviewers' comments:

Reviewer's Responses to Questions

**Comments to the Author**

1. Is the manuscript technically sound, and do the data support the conclusions?

Reviewer #1: No

Reviewer #2: Partly

2. Has the statistical analysis been performed appropriately and rigorously? 

Reviewer #1: No

Reviewer #2: No

3. Have the authors made all data underlying the findings in their manuscript fully available?

Reviewer #1: Yes

Reviewer #2: Yes

4. Is the manuscript presented in an intelligible fashion and written in standard English?

Reviewer #1: Yes

Reviewer #2: Yes

5. Review Comments to the Author

Reviewer #1: Limitations:

- The conclusion is inappropriate and even dangerous because so many variables were not included in the data analysis such as co-morbidity, PH prevention measures such as quarantine, wearing facial masks and social distancing, lock downs, and spread of COVID through the different health regions during the pandemic.

- Another major issue is that you do not include hospital distribution in these health regions. Hospital distribution and the number of hospital beds or ICU beds or “Covid ward” is a function of population density and therefore this should be checked for multi-collinearity with variance inflation factor (VIF). I would argue the same is true for co-morbidities, age, socio-economic status (not able to work from home) instead of just median age.

- I am also not sure if COVID patients were always hospitalized in their own health region. You mention that briefly but maybe you would be able to quantify that and also how that changed during the pandemic. We had many patients’ transfers in the height of the pandemic to neighboring regional hospitals.

- What about the impact of nursing home or long-term care outbreaks, clearly a population who is at highest risk for hospitalization (and death)?

- The hospitalization rate is inappropriate. You are looking at persons hospitalized and not hospital admissions, correct? Therefore, the percentage of person hospitalized instead of rate should be a better measure.

- I think also time is another important variable to be included. We experienced in Louisiana a first wave of almost close to capacity hospitalizations in New Orleans but then the pandemic moved quickly into more rural parts of the state where it most likely spread through households and worksites or houses of religious worship (based on outbreaks identified)

Minor limitations:

- Cross-sectional study OR is really a prevalence odds ratio (POR)

- Table 1: do not use the term “cases” here since this is confusing, you talk about COVID hospitalizations.

- Reference 14 should be removed. Southwest Louisiana was impacted by two major hurricanes in the midst of the pandemic and experienced a surge in COVID cases, no hospital data available because no hospitals were left. Because of the damage and impact of these 2 storms residents had to evacuate to their extended family or hotels and live in crowded homes so their exposure risk (no social distancing or wearing face masks) increased dramatically plus these evacuees were not inclined to get tested because they had other major issues to deal with..

Reviewer #2: I believe this is an important analysis, and authors have done an admiring job in attempting to address the concerns of Reviewer #1. However, some issues remain:

1-Probably the main deficiency of the present analysis is to treat daily hospitalizations as independent data. However, our knowledge of epidemiology tells us that hospitalizations should track the infections in the community, which is a multiplicative, not additive process. Therefore, the fact that one region has more hospitalization rate than another, **could** simply be due to the size of the initial cluster and subsequent hospitalizations may not provide independent information on this. Indeed, the epidemic curves in Fig. 3 clearly show smooth curves, confirming this possibility. Also, different regions may have epidemics starting at different times, thus comparisons on the same day are not appropriate. One possibility is to look at the total hospitalizations around the peak, say until you reach half the peak height on either side, e.g., in Fig. 3.

2-Two important co-variates that are missing from this analysis are the area of the region, and the level of lockdown (e.g. measured by Google Mobility). It is widely believed that the latter is responsible for stopping the first wave across the world, and thus its level at different regions (e.g. two weeks prior to the peak) is an important co-variate. As to the former, the idea that density is the appropriate quantity to correlate is based on assumption of a uniform 2d distribution. The real population distributions, however, are far from uniform, with larger density near metro areas that slowly falls off into the suburbs and rural areas. Therefore, a notion as lived density or population-weighted density (e.g., https://arxiv.org/abs/2007.00159) would be a more appropriate quantifier of local residential conditions. In lieu of this data, a joint correlation with density, area, and mobility should provide a minimal description of underlying dynamics.

6. PLOS authors have the option to publish the peer review history of their article (what does this mean?). If published, this will include your full peer review and any attached files.

Reviewer #1: **Yes: **SUSANNE STRAIF-BOURGEOIS

Reviewer #2: **Yes: **Niayesh Afshordi

---

## [Author Response · Author response to Decision Letter 0]

3 Jun 2021

Response to reviewers

Reviewer #1: Limitations:

- The conclusion is inappropriate and even dangerous because so many variables were not included in the data analysis such as co-morbidity, PH prevention measures such as quarantine, wearing facial masks and social distancing, lock downs, and spread of COVID through the different health regions during the pandemic.

We disagree. The fact that we did not include co-morbidity, PH prevention measures such as quarantine, wearing facial masks, social distancing and lock downs, does not alter the fact that there were large differences in COVID-19 hospitalisation rates across the 11 mainland Scottish health boards, and that these were significantly related to population density. The spread of COVID through the different health regions during the pandemic is what we were studying! Given that lockdown was originally introduced to prevent the NHS from being overwhelmed, we believe our results do indeed support a regional rather than a national approach to lifting or reimposing more restrictive measures (this is currently happening in the UK), and that hospitalisation rates should be part of the decision making process (Scotland reports numbers of COVID-19 patients who are in hospital each day by Health Board but it would make much more sense to express these numbers as rates/100,000).

Another major issue is that you do not include hospital distribution in these health regions. Hospital distribution and the number of hospital beds or ICU beds or “Covid ward” is a function of population density and therefore this should be checked for multi-collinearity with variance inflation factor (VIF). I would argue the same is true for co-morbidities, age, socio-economic status (not able to work from home) instead of just median age.

We disagree. Hospital distribution and the number of hospital beds is indeed a function of population density in Scotland where provision is universal, free at point of use (publicly funded) and adequate. Data released by the Scottish Government confirm there was sufficient hospital bed capacity across Scotland during the first wave of the pandemic (Ref 18).

- I am also not sure if COVID patients were always hospitalized in their own health region. You mention that briefly but maybe you would be able to quantify that and also how that changed during the pandemic. We had many patients’ transfers in the height of the pandemic to neighboring regional hospitals.

Not relevant to Scotland. COVID-19 patients were always hospitalized in their own health region. There were no patient transfers to neighbouring regional hospitals during the height of the pandemic 

What about the impact of nursing home or long-term care outbreaks, clearly a population who is at highest risk for hospitalization (and death)?

We have reported total hospitalisations for a heterogeneous population which includes nursing home residents. The variable of interest in our study was rurality (population density) not the impact of nursing homes. That would have been a different study.

The hospitalization rate is inappropriate. You are looking at persons hospitalized and not hospital admissions, correct? Therefore, the percentage of person hospitalized instead of rate should be a better measure.

Why is percentage of persons hospitalized a better measure than hospitalisation per 100,000? 300/100,000 is 0.3% after all...

I think also time is another important variable to be included. We experienced in Louisiana a first wave of almost close to capacity hospitalizations in New Orleans but then the pandemic moved quickly into more rural parts of the state where it most likely spread through households and worksites or houses of religious worship (based on outbreaks identified)

We did include time as a variable! Our analysis is of hospitalisation across Scotland during the whole of the first wave of the pandemic.

Minor limitations:

- Cross-sectional study OR is really a prevalence odds ratio (POR)

We now describe the study as a repeated point prevalence study in the methods section 

- Table 1: do not use the term “cases” here since this is confusing, you talk about COVID hospitalizations.

Done

- Reference 14 should be removed. Southwest Louisiana was impacted by two major hurricanes in the midst of the pandemic and experienced a surge in COVID cases, no hospital data available because no hospitals were left. Because of the damage and impact of these 2 storms residents had to evacuate to their extended family or hotels and live in crowded homes so their exposure risk (no social distancing or wearing face masks) increased dramatically plus these evacuees were not inclined to get tested because they had other major issues to deal with..

Our reviewer has made some interesting points about what sounds like a very difficult situation in Louisiana. Our paper is, in fact, about rurality in Scotland. We believe the observation that rural communities may behave distinctively in times of natural disaster by expressing greater social control (reference 19) is relevant to our findings.

Reviewer #2: I believe this is an important analysis, and authors have done an admiring job in attempting to address the concerns of Reviewer #1. However, some issues remain:

1-Probably the main deficiency of the present analysis is to treat daily hospitalizations as independent data. However, our knowledge of epidemiology tells us that hospitalizations should track the infections in the community, which is a multiplicative, not additive process. Therefore, the fact that one region has more hospitalization rate than another, **could** simply be due to the size of the initial cluster and subsequent hospitalizations may not provide independent information on this. Indeed, the epidemic curves in Fig. 3 clearly show smooth curves, confirming this possibility. Also, different regions may have epidemics starting at different times, thus comparisons on the same day are not appropriate. One possibility is to look at the total hospitalizations around the peak, say until you reach half the peak height on either side, e.g., in Fig. 3.

We accept all of these points which we feel we have addressed in our discussion when we say ‘An additional measure could be hospitalisation rates within a region, which allow daily small area monitoring of disease spread, albeit with a longer delay caused by the lag between transmission, onset of clinical disease and deterioration to the point of requiring hospitalisation.’ (Discussion, paragraph 3). We go on to argue that ‘other high income countries, including Japan, Germany, South Korea and Hong Kong, have adopted a regional rather than a national approach to lifting or reimposing restrictions based on the number of new cases/100,000 per week.’ The point our paper is attempting to make is that hospitalizations/100,000 could potentially be a more useful metric to consider when making these decisions, given that one of the main purposes of lockdown has been to prevent the health service from being overwhelmed.

2-Two important co-variates that are missing from this analysis are the area of the region, and the level of lockdown (e.g. measured by Google Mobility). It is widely believed that the latter is responsible for stopping the first wave across the world, and thus its level at different regions (e.g. two weeks prior to the peak) is an important co-variate. As to the former, the idea that density is the appropriate quantity to correlate is based on assumption of a uniform 2d distribution. The real population distributions, however, are far from uniform, with larger density near metro areas that slowly falls off into the suburbs and rural areas. Therefore, a notion as lived density or population-weighted density (e.g., https://arxiv.org/abs/2007.00159) would be a more appropriate quantifier of local residential conditions. In lieu of this data, a joint correlation with density, area, and mobility should provide a minimal description of underlying dynamics.

These are undoubtedly important points. It seems highly likely that had we been able to take our analyses to the next level by calculating population weighted density we would have seen an even greater correlation with hospitalisation rates. As it was we found a statistically significant relationship between hospitalisation rates and population density on 97.7% of individual days during the first wave of the pandemic without taking into account the larger population density of inner city areas that slowly falls off into the suburbs and rural areas. To restate the aim of our paper, this is not about the factors that might or might not influence the incidence or prevalence of COVID-19 in different parts of Scotland. We have simply observed that there were huge differences in hospitalisation rates across Scotland during the first wave of the pandemic and shown that these differences were related, at least in part, to population density. Given that lockdown was originally introduced to prevent the NHS from being overwhelmed, we believe our results support a regional rather than a national approach to lifting or reimposing more restrictive measures, and that hospitalisation rates should be part of the decision making process.

6. PLOS authors have the option to publish the peer review history of their article (what does this mean?). If published, this will include your full peer review and any attached files.

Do you want your identity to be public for this peer review? For information about this choice, including consent withdrawal, please see our Privacy Policy.

Reviewer #1: Yes: SUSANNE STRAIF-BOURGEOIS

Reviewer #2: Yes: Niayesh Afshordi

---

## [Decision Letter · Decision Letter 1]

10 Jun 2021

Regional variation in COVID-19 positive hospitalisation across Scotland during the first wave of the pandemic and its relation to population density: a cross-sectional observation study.

PONE-D-20-38210R1

Dear Prof Isles,

We’re pleased to inform you that your manuscript has been judged scientifically suitable for publication and will be formally accepted for publication once it meets all outstanding technical requirements.

Kind regards,

Karyn Morrissey

Academic Editor

PLOS ONE

Additional Editor Comments (optional):

Reviewers' comments:

Reviewer's Responses to Questions

**Comments to the Author**

1. If the authors have adequately addressed your comments raised in a previous round of review and you feel that this manuscript is now acceptable for publication, you may indicate that here to bypass the “Comments to the Author” section, enter your conflict of interest statement in the “Confidential to Editor” section, and submit your "Accept" recommendation.

Reviewer #1: All comments have been addressed

2. Is the manuscript technically sound, and do the data support the conclusions?

Reviewer #1: Yes

3. Has the statistical analysis been performed appropriately and rigorously? 

Reviewer #1: Yes

4. Have the authors made all data underlying the findings in their manuscript fully available?

Reviewer #1: Yes

5. Is the manuscript presented in an intelligible fashion and written in standard English?

Reviewer #1: Yes

6. Review Comments to the Author

Reviewer #1: Thanks for addressing all of my comments even though we still agree to disagree on some issues. :)

I like that your data analysis was done with publicly available PH data! I do hope that this is an ongoing trend so that researchers can utilize more these types of data.

As you stated, the main goal of your manuscript is to look at and decided based on regional and not national data when implementing PH measures during pandemics and epidemics. I think this is a valid and important message which is supported by your manuscript and therefore should be published.

7. PLOS authors have the option to publish the peer review history of their article (what does this mean?). If published, this will include your full peer review and any attached files.

Reviewer #1: **Yes: **SUSANNE STRAIF-BOURGEOIS

---

## [Editor Report · Acceptance letter]

30 Jun 2021

PONE-D-20-38210R1 

Regional variation in COVID-19 positive hospitalisation across Scotland during the first wave of the pandemic and its relation to population density: a cross-sectional observation study. 

Dear Dr. Isles:

I'm pleased to inform you that your manuscript has been deemed suitable for publication in PLOS ONE. Congratulations! Your manuscript is now with our production department. 

Kind regards, 

on behalf of

Dr. Karyn Morrissey 

Academic Editor

PLOS ONE